# Reversible and spatiotemporal control of colloidal structure formation

H. Dehne[1], A. Reitenbach[1] & A. R. Bausch [1✉]

Tuning colloidal structure formation is a powerful approach to building functional materials, as a wide range of optical and viscoelastic properties can be accessed by the choice of individual building blocks and their interactions. Precise control is achieved by DNA specificity, depletion forces, or geometric constraints and results in a variety of complex structures. Due to the lack of control and reversibility of the interactions, an autonomous oscillating system on a mesoscale without external driving was not feasible until now. Here, we show that tunable DNA reaction circuits controlling linker strand concentrations can drive the dynamic and fully reversible assembly of DNA-functionalized micron-sized particles. The versatility of this approach is demonstrated by programming colloidal interactions in sequential and spatial order to obtain an oscillatory structure formation process on a mesoscopic scale. The experimental results represent an approach for the development of active materials by using DNA reaction networks to scale up the dynamic control of colloidal self-organization.

[1] Center for Protein Assemblies (CPA) and Lehrstuhl für Biophysik (E27), Physics Departement, Technische Universität München, D-85748 Garching, Germany. ✉email: abausch@mytum.de

Guiding the self-organization of colloidal particles is a powerful concept for building materials that can adapt their optical, mechanical, magnetic, or electrical properties[1–3]. The positional order of the colloids is decisive for the final material properties and gives rise to a myriad of functional applications with unique features, as shown in photonics, electronic materials, lithographic masks, and catalytic scaffolds[4–9]. One essential advantage of colloidal systems is the ability of controllable self-assembly tailored by the interactions of the individual particles[10]. This enables the controlled formation of structures such as dispersed solutions, clusters, amorphous gel-like structures, and different types of colloidal crystals or even colloidal machines[11–14]. To date, external stimuli such as temperature shifts or defined chemical gradients are mostly used to drive the dynamics of structure formation[15,16]. DNA hybridization has proven to be best-suited method to control the interactions of DNA-coated colloids due to its programmable binding strength and selectivity[15,17–23]. The use of enzymatic reaction networks to control DNA strand production introduces the ability to control colloidal aggregation in time and space. By this approach, cellular properties such as communication and signaling cascades can be mimicked[24]. Autonomously acting colloidal systems that can perform transient aggregation pulses with tunable steady states have been realized[25–27] by relying on the consumption of molecular fuel without further reversibility. Simply harnessing the supply of fuel is not sufficient to realize dynamic colloidal systems that are capable of autonomous cycles or oscillations[28,29]. Such systems require the coupling of enzymatic reaction networks far from thermodynamic equilibrium with nonlinear dynamics and feedback loops for the aggregation kinetics of colloidal particles without any external driving[30,31].

Here, we harness the enzymatic activity of a polymerase, exonuclease, and nickase (PEN) to construct an enzymatic reaction network, which produces DNA strands that drive the aggregation of micron-sized colloidal particles. Using the programmability of the underlying enzymatic reaction network, we were able to control the colloidal aggregation speed and introduce defined time delays to shape the final colloidal composition of a multicomponent system. We designed reaction networks that realize the spatial propagation of colloidal aggregation fronts and were able to induce a communication mechanism within cm-long channels. Finally, an enzymatic predator-prey reaction network was used to drive autonomous oscillatory colloidal structure formation without the need for external stimuli.

## Results

We engineered a series of dNTP-driven enzymatic reaction networks based on the PEN toolbox[32,33], a DNA programming framework designed to construct out-of-equilibrium systems performing DNA amplification (polymerase and nickase, Fig. S1) and DNA degradation (exonuclease, Fig. S2)[34–36].

Enzymatic reactions are used to produce DNA linker strands, which are able to induce the specific structure formation of DNA-coated colloids. We functionalized 1 µm particles with three different DNA strands ($\bar{\alpha}, \bar{\beta}$, and $\bar{\gamma}$) and controlled their specific interactions using the complementary linker strands (αβ and αγ), which are able to bind the respective colloids with each other (Fig. 1a, b). The activation of the enzymatic reactions is realized by the binding of primer strands (δ or ε) to a complementary DNA template (e.g. $\overline{\delta to\alpha\beta}$). The nomenclature of the DNA template strands was chosen to be a combination of the name of the primer (input) and the name of the resulting product (output). For instance, the strand $\overline{\delta to\alpha\beta}$ is activated by a δ strand and produces the linker strand αβ, which induces the aggregation of colloids functionalized with $\bar{\alpha}$ and $\bar{\beta}$ DNA. The primer strand

concentration δ or ε determines the linker production rate and thus the speed of colloidal aggregation. The network kinetics are therefore controlled by enzymatic reactions, which amplify or degrade the primer strands.

The enzymatic reactions are designed modularly and can be categorized into three groups: (1) Self-replication: A first-order positive feedback loop uses the DNA template $\overline{\delta to\delta}$ or $\overline{\epsilon to\epsilon}$ to amplify the 10 nT primer strand δ or ε, respectively (δ = CATTCGGCCG, ε = CATTCAGACG). The weak interaction between primer and template strands prevents the saturation of the formed complexes during primer production and facilitates nonlinear production (Fig. 1c). (2) Inhibiting reactions: Three different reactions can be used to control the amplification of the primer strands by degradation, by deactivation of the primer activity, or by a negative feedback mechanism (Fig. 1d). (3) Communication: Each primer can induce the production of the other strand (Fig. 1b).

**Controlling the colloidal composition starting from homogenous conditions.** In the first step, we established enzymatic control of the colloidal interactions and their structure formation. Each of the self-replicating reaction networks $\overline{\delta to\delta}$ and $\overline{\epsilon to\epsilon}$ can exclusively prime the corresponding linker production $\overline{\delta to\alpha\beta}$ and $\overline{\epsilon to\alpha\gamma}$ and induce selective colloidal cluster aggregation (Figs. 2a, b and S3). The colloidal assembly is characterized by fractal growth of the particles due to the strong interactions of the complementary linker and colloidal DNA strands. The colloids used in the reactions are fluorescently barcoded and can thus be distinguished ($\bar{\alpha}, \bar{\beta}$ = green and $\bar{\gamma}$ = red). The aggregation induced by the αβ linkers leads to homogenously green colloidal clusters, while the αγ linkers form heterogeneous clusters; only red and green particles can aggregate with each other (Fig. 2a, b, confocal image). The speed of aggregation is controlled by tuning the enzymatic reaction network. Here, different concentrations of $\overline{\delta to\delta}$ are used to control the speed of linker production, effectively guiding the time traces of the aggregation (Fig. 2a).

In the next step, the two self-replicating networks were used simultaneously and coupled via the $\overline{\delta to\epsilon}$ communication strand, while the reaction was initiated by using the δ primer strand (Fig. 2c). As a consequence, we introduced a time delay, where the colloidal aggregation of the green colloids is induced at first, followed by the red particles enclosing the green clusters acting as an alloy. Since the time delay is given by the delayed initiation of the $\overline{\epsilon to\epsilon}$ reaction, we can adjust the time traces of the red colloidal clustering simply by varying the concentration of $\overline{\delta to\epsilon}$ (Fig. 2c). In the experiment, the time delay of the αγ linker production and the corresponding colloidal clustering was controlled within a period of one hour. The experimental observations were reproduced by a simulation of two coupled autocatalytic reactions (Fig. 2c) (code attached in Supplemental Materials).

**Molecular communication and traveling aggregation fronts.** To achieve local activity within distinct compartments, we tethered the DNA linker-producing strands to 2 µm-sized colloids (e.g. $P_{\overline{\delta to\alpha\beta}}$) and thus spatially constrained linker production to these particles, while the products were free to diffuse. Colloidal cluster aggregation was observed in a microscopic chamber with two reservoirs connected by an ~1 cm long channel (Fig. 3a). Here, the $P_{\overline{\delta to\alpha\beta}}$ particles were introduced only into one reservoir (red dotted square), while the reaction buffer including all enzymes and primers was homogenously distributed in the chamber. As expected, colloidal aggregation takes place only in the reservoir with the tethered colloids. Shortly after the reaction starts, the aggregation patterns are defined by the random position of the

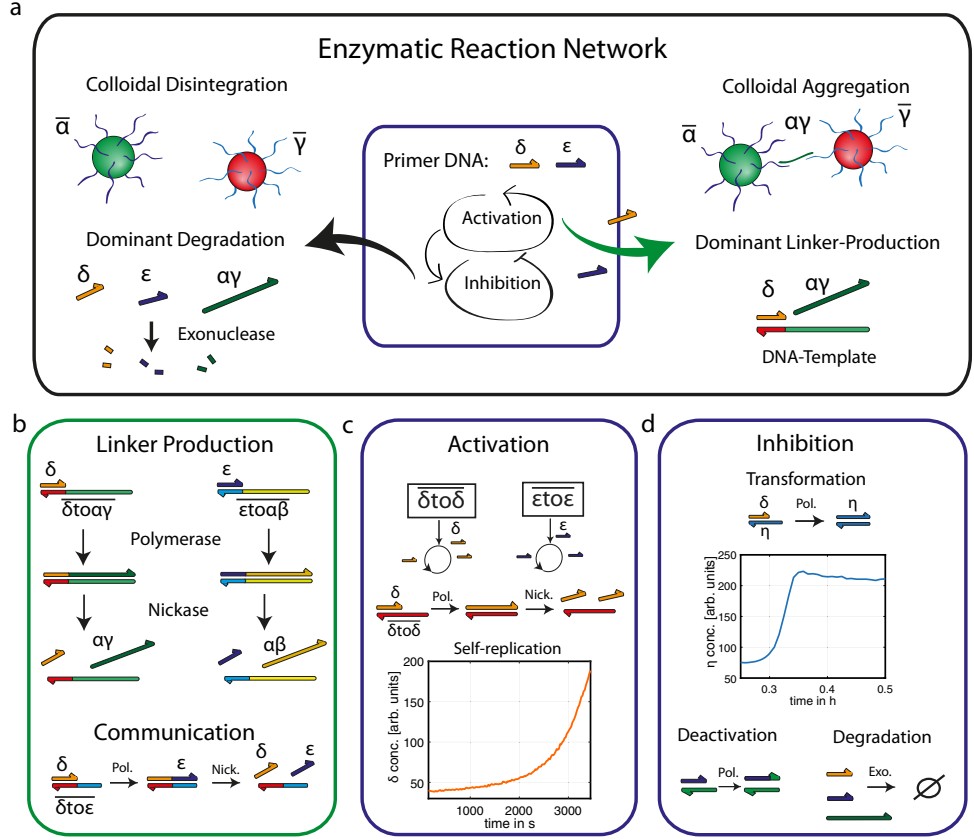

**Fig. 1 DNA reaction networks control colloidal aggregation. a** Scheme of an enzymatic reaction network that can inhibit and activate the amplification of the primer DNA strands ε and δ. In the presence of the primer strands, an additional enzymatic reaction is activated and produces a DNA linker strand, which induces the selective structure formation of DNA-coated colloids. **b** The production of two different types of colloidal linker strands αγ and αβ (24 nT) can be activated by the primer strands ε and δ (10 nT) and results in the specific aggregation of DNA coated colloids, which are fluorescently labeled. In the communication setup, each of the primers is used as a catalytic input to produce the other primer as an output. **c** The autocatalytic DNA amplification is catalyzed by repetitive DNA templates using a polymerase (Pol.) and a nickase (Nick.) and is monitored using fluorescence spectroscopy (here, $\overline{\delta to\delta}$ = 120 nM). **d** The self-replication of the primer strands (ε and δ) can be controlled by three inhibiting reactions. A palindromic DNA sequence η is used to realize the autocatalytic transformation of δ into η (experiment: η = 50 nM and δ = 1 nM). The deactivation of the primer activity is achieved by the addition of a short polynucleotide sequence, and an exonuclease (Exo.) is used to degrade the produced DNA strands.

linker producing colloids (red mask), caused by the local concentration gradients of DNA linkers.

In the next step, we aimed to create the propagation of "materialization", initiated by a local event. Therefore, we used the template $\overline{\delta to\delta}$ in the buffer and activated the reaction only locally by the addition of the primer strand δ into only one reservoir. Consequently, a concentration gradient formed, leading to diffusion of the production of δ into the channel. This setup generates a reaction-driven propagation of δ, where the speed is defined by the ratio of the enzymatic reaction rate and the diffusive transport[37]. The modular enzymatic setup allows us to combine the reaction-driven propagation of δ with colloidal aggregation by using $\overline{\delta to\alpha\beta}$ strands that are free in solution. As expected, the activation of δ induces local aggregation at the reservoir with the subsequent propagation of an aggregation front along the channel (Fig. 3b). The front propagates over the complete distance and exceeds the speed and distances possible for a purely passive diffusive process. Since the propagation of δ is reaction driven, varying the concentration of $\overline{\delta to\delta}$ allows tuning of the reaction speed (Fig. 3b, inset). The distance of propagated aggregation was monitored after 90 min for different template concentrations and showed a linear dependence on the initial concentration of $\overline{\delta to\delta}$.

In addition to the direct propagation of aggregation, the system can also be used to realize communication between different compartments using signaling cascades. Therefore, we added $\overline{\delta to\delta}$ to the bulk solution and started the reaction at one reservoir by the addition of δ, while the linker-producing particles $P_{\overline{\delta to\alpha\beta}}$ were placed on the opposite site. As a result, the δ strands induced a traveling front along the channel and only induced colloidal aggregation only once the wave reached the $P_{\overline{\delta to\alpha\beta}}$ particles. This can be seen by the absence of colloidal aggregation in the reservoir where the primer strands δ were introduced and the delayed aggregation at the end of the channel, which starts after approximately one hour (Fig. 3c).

We expanded the reaction network to create a two-level communication system that can send and return information[24]. To this end, we used both excitable templates $\overline{\delta to\delta}$ and $\overline{\epsilon to\epsilon}$ in the solution and placed transformation particles $P_{\overline{\epsilon to\delta}}$ into only one reservoir. The reaction was triggered by introducing the primer strand ε at the opposite site of the channel. Figure 3d sequentially shows the reaction start using the ε primer at one site of the channel and the propagation of ε along the channel mediated by the $\overline{\epsilon to\epsilon}$ templates. The primer strand ε is transformed locally into δ using $\overline{\epsilon to\delta}$, which then triggers the backward propagation of the δ primer. Due to the linker-producing $\overline{\delta to\alpha\beta}$ strands, which are free in solution, a backward aggregation of colloids is induced, which starts at the end of the channel and subsequently propagates in backward.

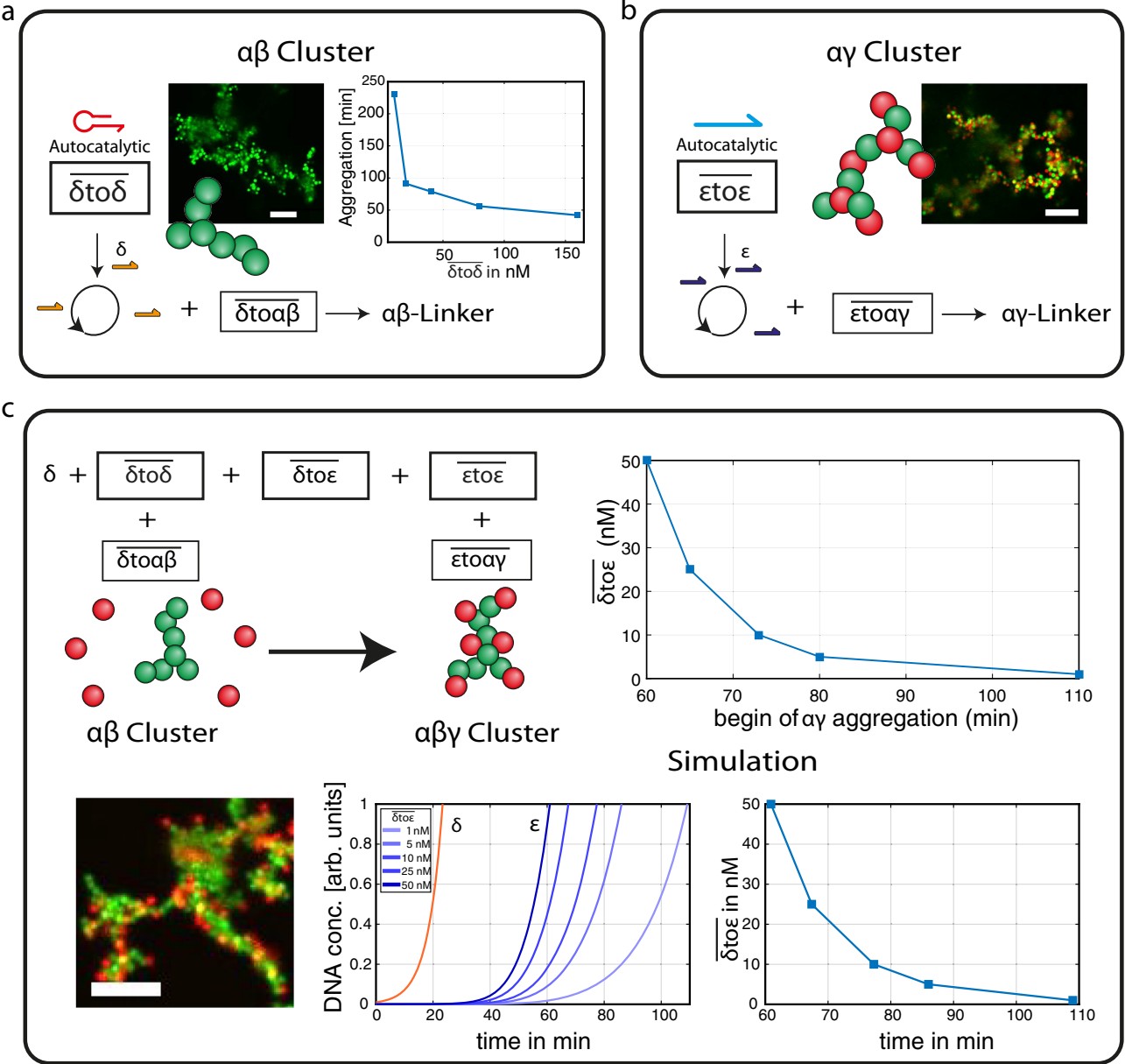

**Fig. 2 Enzymatic reactions control the speed and composition of colloidal aggregation. a** The aggregation of $\bar{\alpha}$ and $\bar{\beta}$ particles (green) is induced by enzymatic reactions, and the aggregation speed can be tuned by using different template concentrations ($\overline{\delta to\delta}$). **b** The autocatalytic production of $\varepsilon$ is used to control the structure formation of $\bar{\alpha}$ and $\bar{\gamma}$ particles and results in clusters of red and green particles. **c** The two self-replicating reactions can be coupled by using the "communication-reaction" of $\overline{\delta to\varepsilon}$. The amplification of $\delta$ is induced at first, and the time-delay of the subsequent reaction is determined by the concentration of $\overline{\delta to\varepsilon}$. The delayed production of $\alpha\gamma$ linker results in a controllable initiation of $\bar{\alpha}$ and $\bar{\gamma}$ aggregation. The programming of the enzymatic setup allows tuning of the colloidal composition, as shown by the green backbone ($\bar{\alpha}$ and $\bar{\beta}$) and the surrounding of red particles ($\bar{\gamma}$, scale bar = 10 μm). The time traces of the two reactions are simulated to demonstrate the induced time-delay. The amplification of $\delta$ is kept constant, while the concentrations of $\overline{\delta to\varepsilon}$ are varied to control the initiation of the $\varepsilon$ amplification. The time of aggregation was determined at $\varepsilon = 1$ arb. units.

**Oscillatory structure formation**. An oscillatory colloidal structure formation process is realized on the basis of a previously introduced molecular predator-prey reaction network[38], which periodically produces and degrades the primer strand δ. The nomenclature of the enzymatic reaction network used is analogous to the predator–prey population dynamics described by the Lotka-Volterra equations[39]. The oscillation of the DNA strand concentrations is based on three enzymatic reactions: exponential DNA amplification (δ = prey), a negative feedback loop (η = predator), and a degradation reaction (natural decay). The reaction conditions were optimized to obtain an oscillating regime of the predator-prey system facilitating increased DNA

concentrations and longer DNA cycles with lifetimes of up to 5 h with sustained oscillation over 60 h.

The characteristics of the oscillations are based on the interplay of the single enzymatic reactions, which depend on the temperature, the DNA interactions, and the enzymatic activity. One way to control the kinetics of the oscillation is the absolute concentration of $\overline{\delta to\delta}$ strands, since the template strands directly affect the autocatalytic production. Thus, the frequency of the oscillating network can be systematically tuned by the concentration of this primer strand production template $\overline{\delta to\delta}$ (Figs. 4a and S4).

To exploit the oscillation of the DNA primer strands and control colloidal aggregation on the mesoscale, it is necessary to

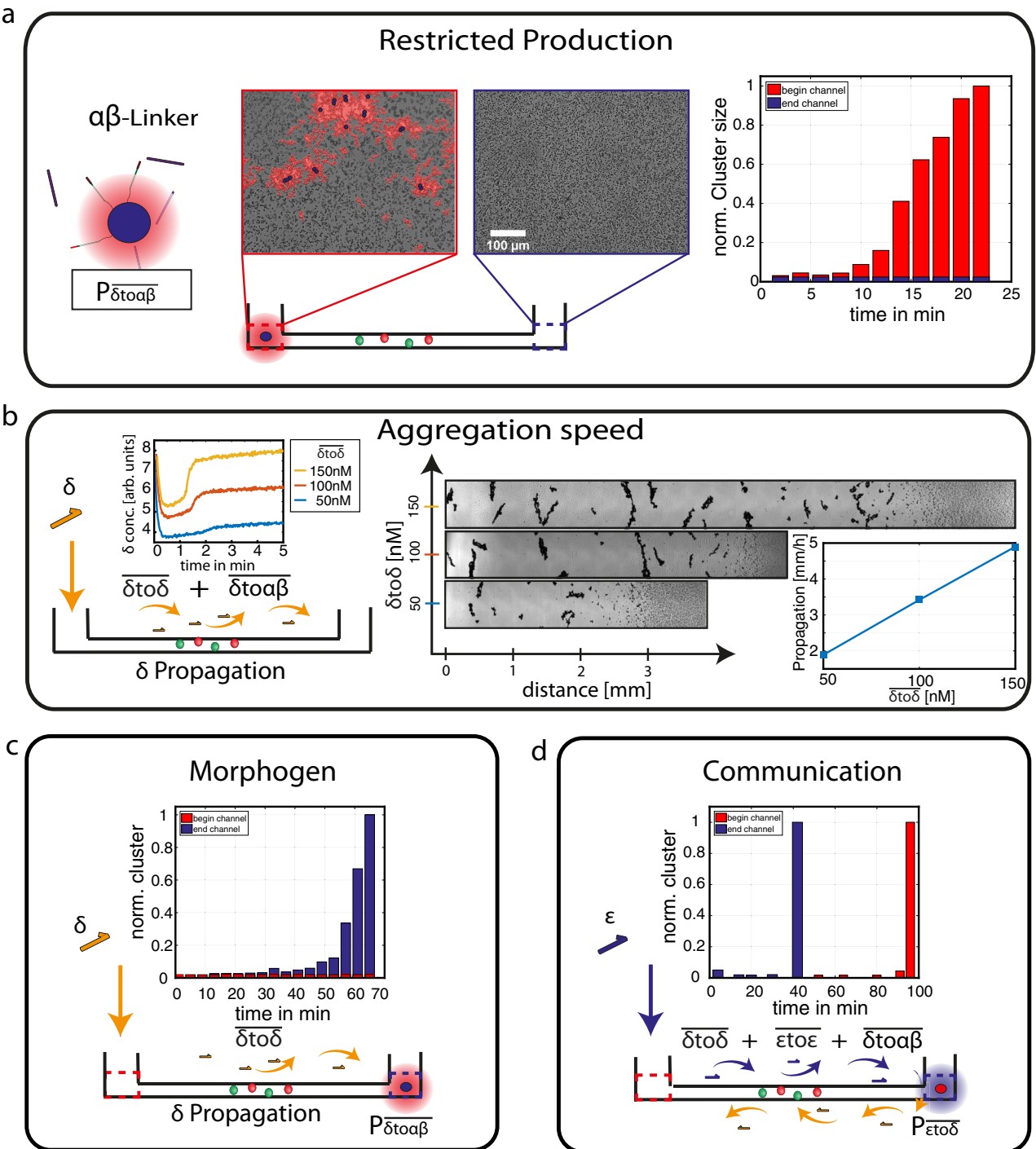

**Fig. 3 Traveling aggregation fronts and communication. a** The local linker production of functionalized particles ($P_{\overline{\delta to\alpha\beta}}$) generates concentration profiles and results in local aggregation (red area) within a microfluidic chamber (~1 cm channel). **b** Traveling fronts of colloidal aggregation are induced by the local activation and subsequent propagation of the primer strands. The speed of the propagation is tuned by varying the reaction speed by the global concentration of $\overline{\delta to\delta}$ strands. The images of the colloidal aggregates along the channel are depicted at $t = 90$ min. **c** Signal propagation along the microfluidic channel is realized by the local activation at one microfluidic chamber and the propagation of δ along the channel. The local colloidal structure formation is induced by the activation of the linker production once the signal reaches the functionalized particles ($P_{\overline{\delta to\alpha\beta}}$). **d** Both self-replicating systems are used to enable the communication between the two microfluidic chambers. The amplification of ε is activated locally, leading to the signal propagation, which is transformed to δ at the end of the channel due to the functionalized particles $P_{\overline{\epsilon to\delta}}$. The transformation induces a backward signal propagation which is finally coupled to the colloidal aggregation ($\overline{\delta to\alpha\beta}$).

produce linker strands exclusively at the peaks of the amplification cycles, while degradation must dominate within the valleys. This was realized using the already introduced reaction of the linker-producing strand $\overline{\delta to\alpha\beta}$ (Fig. 4a, red-dotted box). The

strand was designed to exclude destructive interferences with the predator-prey system. Furthermore, the interactions of the primer and linker strands were optimized to realize the switch between colloidal aggregation and disintegration (Fig. S5–7). The

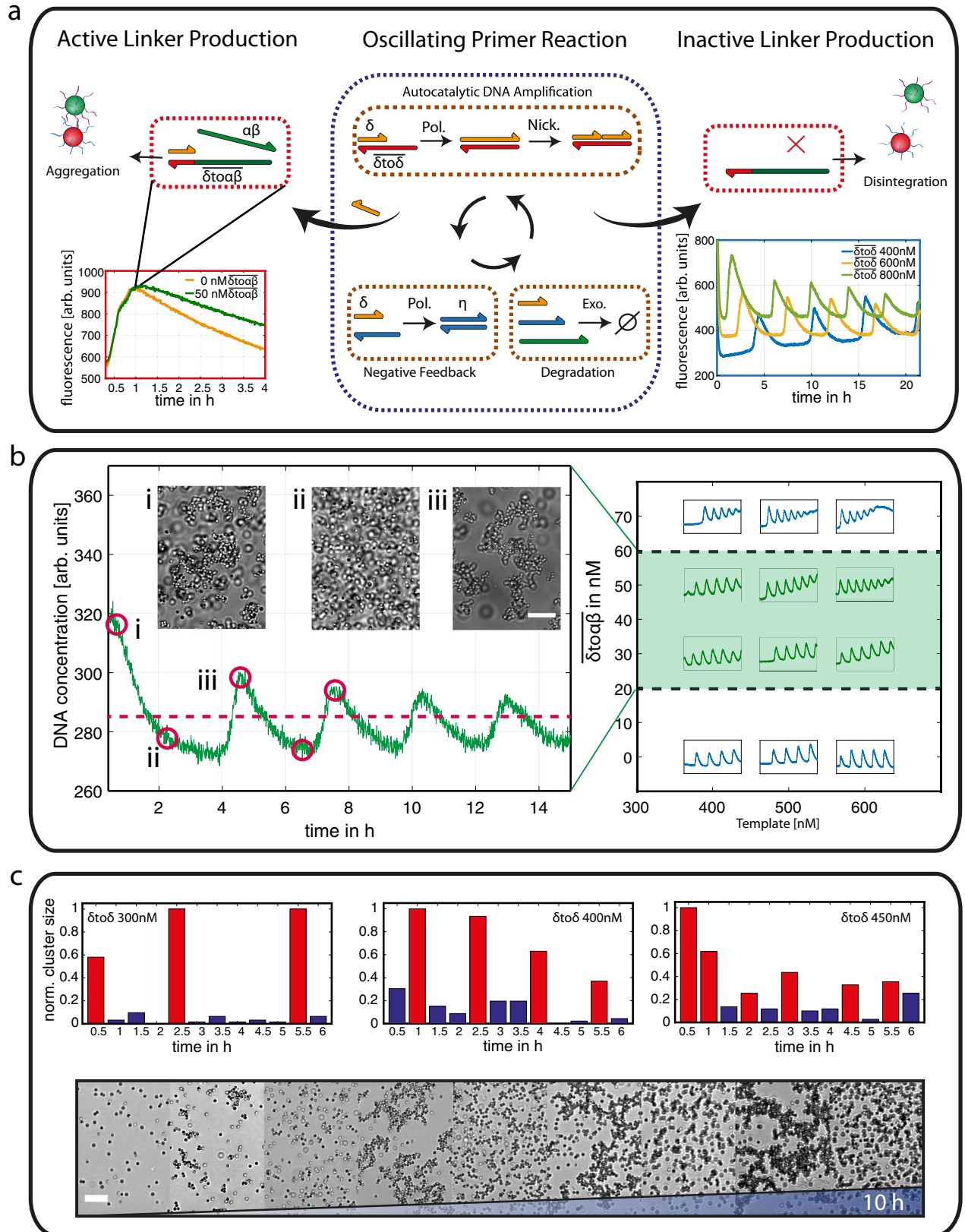

activation of linker production during one cycle of oscillation is demonstrated in Fig. 4a.

We analyzed the dependence of the phase space of the oscillating regime on the concentration of the template $\overline{\delta\text{to}\delta}$ and

linker-production strand $\overline{\delta\text{to}\alpha\beta}$ (Fig. 4b). On the one hand, the limits are given by the stability of the enzymatic reaction. While low concentrations of DNA templates ($\overline{\delta\text{to}\delta} < 200$ nM) lead to long lag phases or the absence of DNA amplification, template

**Fig. 4 Oscillating structure formation. a** Scheme of the oscillating reaction network (blue dotted box) and the coupled linker reaction (red dotted boxes). The oscillation of the DNA amplification (blue box) can be controlled in terms of frequency and can activate the linker production during the DNA concentration peaks (red box). Here, one cycle of the oscillation is shown with (green) and without (orange) additional linker production. **b** Oscillating linker concentrations in dependency of the template concentration and linker production. The green area depicts the phase space, which enables the synchronization of the DNA concentration profiles and the colloidal aggregation. At different points in time, colloids were added (red circles) and the microscopy snapshots demonstrate the corresponding aggregation (i, iii) and disintegration (ii), scale bar = 15 μm. **c** Colloidal cluster size monitored over time for different template concentrations (see "Methods", microscopy experiments). The monodisperse state is depicted in blue, while the colloidal clusters are highlighted in red. Bottom: Microscopy snapshots of the colloidal structure formation during 10 h of oscillating linker production (Supplementary Video, $\overline{\delta to \delta}$ = 400 nM, with peaks of aggregations at 51, 149, 243 and 557 min, scalebar = 15 μm).

concentrations exceeding a certain limit (>600 nM) result in damped oscillations (Fig. S4). On the other hand, the linker strands need to be within a concentration regime where structure formation can readily switch between colloidal aggregation and disintegration. This was experimentally determined by the addition of colloids at different points in time (Fig. 4b, red circles) during the observation of oscillating linker production. Within a regime of $\overline{\delta to \alpha\beta}$ = 20–70 nM (Fig. 4b, green area), colloidal structure formation follows the exact same time trace as the oscillating DNA profile. Finally, we analyzed colloidal structure formation in an experimental setup in which colloids and the reaction network were present in bulk from the beginning and observed oscillations of up to 5 cycles with different frequencies (Fig. 4c and Supplementary Video).

## Discussion

The functionality and complexity of biological systems rely on the self-organization of multiple components, which are orchestrated by biochemical reaction networks. The interplay of nonlinear reactions facilitates the formation of activation patterns, which enable a variety of dynamic structures across many length scales to perform biochemical and mechanical tasks. From a materials science point of view, guiding the self-organization of synthetic building blocks by using such reaction networks is therefore a powerful concept, as there are many fascinating properties of biological systems that would be of interest to mimic.

To this end, we developed a versatile and tunable enzymatic reaction network and introduced additional translational steps to couple the molecular programmability to the material properties of a colloidal system on the mesoscale. In the first step, we used two different enzymatic reaction networks to amplify distinct DNA linker strands, which are able to induce the selective structure formation of DNA-coated colloids. By using transformation reactions, we introduced time delays between the two linker-producing networks. The programmable time delays were used to tune the time traces of the colloidal structure formation and control the final colloidal composition within a multi-component colloidal system. In the next step, we utilized the autocatalytic amplification reaction to realize spatial propagation of the primer DNA, which is, in contrast to passive diffusive systems, driven by the speed of the enzymatic reaction. By using linker-producing reactions, it is possible to realize the tunable propagation of colloidal structure formation on a centimeter length scale. Moreover, a localized transformation reaction was harnessed to develop a colloidal system that can send primer DNA as a chemical signal and induce colloidal structure formation and its subsequent propagation at a desired time and place. Finally, we used the autocatalytic inhibition and degradation of the primer strands provided by the molecular predator-prey reaction to achieve autonomous and reversible linker production. Therefore, it was possible to couple the oscillatory dynamics of the primer DNA to colloidal structure formation on the mesoscale.

The coupling of enzymatic reaction networks and mesoscopic systems such as colloidal aggregation requires addressing a range of challenges, such as matching the dynamics of the molecular reactions and the colloidal system as well as disturbing interferences and the competing demands for stability and reversibility of the involved DNA hybridization energies. For instance, autocatalytic amplification and inhibition reactions require low interaction energies of rather short primer DNA strands to prevent saturation of the DNA templates and the concomitant halt of the reaction. However, longer linker strands with higher binding affinities are required to drive the aggregation of micron-sized colloids. To address these divergent demands, a separate linker-producing reaction, which is activated by small primer strands, needs to be introduced. This step must not interfere with the primer pool to reproduce the dynamics of the primer concentration with the production of linker strands. In addition, the linker strands need to be strong enough to induce colloidal aggregation but weak enough to allow their disintegration, which is mediated by the equilibration of the DNA concentration.

Since the length of the linker and the corresponding binding affinity can be adapted to the application, the system has the potential to be transferred to a variety of applications that rely on DNA hybridization. Thus, it may prove to be useful to facilitate the development of materials with dynamic and tunable pattern formation to realize dynamic and autonomously acting materials.

## Methods

**Sample preparation**. All experiments were performed at $T$ = 46.5 °C in a reaction buffer containing 20 mM Tris-HCl, 10 mM (NH4)2SO4, 10 mM KCl, 50 mM NaCl, 8 mM MgSO$_4$, and 400 μM dNTP Mix (*NEB*) at a pH of 8.8. In addition, 0.1% SynperonicF108 (*Sigma Aldrich*), 4 mM DTT, 100 μg/mL BSA (*NEB*), and 2 μM netropsin (*Sigma Aldrich*) were used as stabilizing agents for the enzymatic reactions. The extremely thermostable single-strand binding protein ETSSB (*NEB*) was used at 10 ng/μL to prevent the formation of weak secondary structures of the DNA species. The nicking enzyme Nb.Bsmi and the Bst. Polymerase large fragments were obtained from *New England Biolabs*, and the exonuclease ttRecJ (*Thermus Thermophilus*) was purified in our laboratory. The nickase was used at 600 U/ml and the exonuclease at 4.42 nM in all experiments unless otherwise specified. The polymerase was used at 20 U/ml for the oscillating reactions (Fig. 4) and at 80 U/ml in all experiments, as shown in Figs. 2 and 3. In addition, the T4 gene 32 protein was used at 33 μg/ml to prevent an unintended side reaction (Fig. S8) leading to exponential amplification of arbitrary DNA sequences. This reaction occurs for long time periods in the presence of Bst. Polymerase and the nicking enzyme Nb.Bsmi.

**Colloidal functionalization**. The streptavidin-coated colloids were functionalized with biotinylated DNA to obtain particles with a double-stranded spacer and a single-stranded end, which were monodisperse in solution and aggregated once the complementary linker strand was present. We used 1 μm paramagnetic and streptavidin-coated microspheres (*Dynabeads MyOne C1, Invitrogen*). The colloids were prepared by mixing 25 μL of colloids (10 mg/mL) with 5 μL of biotinylated docking DNA (1.67 μM final concentration) and incubating them for 30 min at room temperature. Afterward, they were washed three times with 100 μL of reaction buffer (20 mM Tris-HCl, 10 mM (NH4)2SO4, 10 mM KCl, 50 mM NaCl, and 8 mM MgSO4) mixed with 0.1% Pluronic F127 (Sigma Aldrich) and 1.67 μM spacer DNA. After another incubation step of 15 minutes at room temperature with continuous mixing, colloids were washed three times in the same buffer and then used for the experiments. They were newly prepared every day. The experiments, which were observed using confocal microscopy, were performed with 1 μm fluorescent polystyrene microspheres (ex/em: 660/690 nm and ex/em: 480/520 nm)

purchased from *Bangs Laboratories*. The microspheres (1% w/v) were functionalized with biotinylated docking DNA for at least 12 h on a rotator at 4 °C, and the washing steps were performed by centrifugation of the samples at 1200 relative centrifugal force and replacement of the supernatant with the buffer solution. The final colloidal solution was sonicated for 30 min to prevent nonspecific aggregation.

**Microscopy experiments**. Bright-field optical images and videos were obtained using a Leica DMI6000B and an HCX PL APO 40×/1.25–0.75 oil CS objective. For confocal microscopy, a Leica TCS SP5 was used with an HCX PL APO 63×/1.40–0.60 oil CS objective. The samples were observed within homemade microscopy chambers made of glass slides and single parafilm layers, which were cut to a sample volume of ~10 µl. The chambers were sealed with vacuum grease to avoid evaporation. The experiments in Fig. 3 were performed using the *ibidi* channel slide *µ-Slide VI 0.1*. Here, 15 µl of mineral oil was used for each reservoir to prevent evaporation. The experiment in Fig. 3d was performed using 10 nM

εpoly t to prevent an autocatalytic start of $\overline{\epsilon\tau\sigma\epsilon}$. The microscopy chambers were heated by a water-tempered copper chamber and objective warmer and calibrated to 46.5 °C. The experiments in Fig. 4c (boxplots) were performed using a thermocycler at constant working temperature and multiple identical samples. At a desired point in time, they were transferred to heated observation chambers, and images were taken after 10 min to observe structure formation.

**Fluorescence spectroscopy**. Fluorescence measurements were performed at 47 °C using a *SpectraMax M5* plate reader (*Molecular Devices*) and *MICROPLATTE 384 WELL, PS, F-BODEN, µCLEAR®* plates by Greiner Bio-One with a sample volume of 15 µl, and an additional 10 µl of mineral oil was used to prevent evaporation during the measurements. To visualize the production and degradation of the oscillating network, the DNA intercalator *EvaGreen* (*Biotium*, ex/em: 500/530 nm) was used undiluted at 1× concentration (stock concentration: 20× in water).

**Data analysis**. The average cluster size of the colloidal aggregates is determined by thresholding the bright field images and excluding single particles to create a binary image using the software tool to analyze particles of ImageJ. The obtained masks of the colloidal aggregates were analyzed in terms of size and total amount.

**Reporting summary**. Further information on research design is available in the Nature Research Reporting Summary linked to this article.

## Data availability
All data generated in this study are provided in the Supplementary Information files.

## Code availability
All codes generated in this study are provided in the Supplementary Information files.

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

## Acknowledgements
We thank Fritz Simmel for helpful discussions. We gratefully acknowledge the financial support of the Deutsche Forschungsgemeinschaft (DFG) through Project ID 201269156–SFB 1032. We gratefully acknowledge funding by the Bavarian Ministry of Science and the Arts through the ONE MUNICH Project "Munich Multiscale Biofabrication".

## Author contributions

A.R.B and H.D. designed the experimental system. H.D. developed the colloidal systems and reaction networks, analyzed the data, and performed simulations. Experiments were performed by H. D and A.R., A.R.B. and H.D. wrote the paper and A.R.B. supervised the research.

## Funding

.

## Competing interests

The authors declare no competing interests.
