## [Peer Review File · Nature Communications]

REVIEWER COMMENTS

Reviewer #1 (Remarks to the Author):

This manuscript describes mesoscopic particle assembly formation using enzymatic activity. The authors show dynamic control over the assembly of colloids by the use of variable DNA linker sequences. Results include oscillating colloidal aggregation over time, which is controllable by oscillating linker production over time. Overall, there is some really exciting science in this manuscript. Nevertheless, I have a real problem with this manuscript. The manuscript is poorly written and organized, and there are lots of typos, that make it hard to read and understand. It also bluntly misrepresents the field. For example the statement: 'Due to the lack of control and reversibility of the interactions, a dynamic and autonomous acting system on a mesoscale was not 10 feasible yet.' is just not true. There are a large number of papers in the literature that have used DNA melting temperature as a key part of a dynamic colloidal system. I understand what the authors want to say but the way they phrased it is misleading. Even colloidal machines (and I guess one can call them dynamic) have now been reported. Overall, this manuscript needs major revisions and clarification in some sections to clearly evince the study's points to readers. The manuscript is so full of jargon and unexplained abbreviations that even an expert on colloidal self-assembly has to re-read this manuscript multiple times to get the point.

- Check for typos. For example, tunable is spelled incorrectly in the abstract, production in figure 1 (german spelling), etc.
- Please re-think Figure 1. It would be great if you can provide a figure that shows the overall concept. Figure 1 is not it and if you think it is, it is WAY too complicated for the non-expert reader.
- The Nomenclature is very confusing and never explained. For example, what is the difference between " $\delta to \delta$ " and " $\delta to \delta$ ", or they are just typos? More general, what is for example δ ? What is the DNA sequence (it is in the supplemental information but one cannot understand the article without it. This should be in the main text or at least the critical parts)? Why were these strands used? In Figure 2, there are spaces between " $\delta to \delta$ " (and other labels) that are not consistent with the rest of the work. The reader suggests, if possible, a change to " $\delta to \delta$ " (and similarly to all other labels) for the sake of readability and clarity. Moreover, the authors used " $\delta to \alpha\beta$ " to indicate linker-production, while use the say way of " $\delta to \delta$ " to represent the template for the production. Please find a clearer way to abbreviate the products or methods and make them consistent both in text and Figures.
- The introduction should be edited to make clearer the background, motivation, and the proposed experimental methods of this work. For example, it is unclear how the work will relate to biological systems, which is the first topic introduced, until enzymatic activity is discussed 2 paragraphs later. It also does not provide a clear background of the state of the art in colloidal science. For example, self-replication has been reported for colloids. Cite it. Communication has been reported for colloidal systems. Same is true for inhibition. Same is true for dynamic switching.
- Citations are missing in the introduction. Some sentences that introduce new concepts lack citations.
- Please always use either "past tense" or "present tense" when describing the experiments in your discussion part (either is fine but be consistent).
- Figure 2. First, there should be a confocal image for illustration of $\alpha\beta$. Second, what does the yellow color indicate, since there are only green and red particles aggregating? Third, both confocal images in b and c are low quality and low magnification that it is not convincing to show the colloidal structure as described by cartoon images. One can actually see green-green interactions and red-red interactions. Quantification is needed.
- In figure 4a, graphs are overlapping and should be moved for readability. For the figures used for explanation of tuning frequency with different concentrated $\delta to \delta$, the ones shown in Figure S4c is less confusing than the current one.
- Figure 4 b, what are those three microscopy images (i, ii, iii)? There are no information or discussion about them.
- Figure 4c, for $\delta to \delta = 400\text{nM}$ group, as shown in the graph above, the cluster size at time 4.5-5 h was much smaller than the very beginning (individual particles). However, the clusters (around that time)

look bigger and concentrated from the microscopy snapshots below.

- The discussion section reads as a conclusion.

Reviewer #2 (Remarks to the Author):

Abstract: Need specifics on what material properties are accessible using the building blocks and interactions. Unclear how DNA reaction circuits control the interactions and reversibility of particle assembly.

Figure 1: a. Figure is quite confusing. Hard to differentiate between how e and d mechanisms differ. B. How does the short polynucleotide sequence result in the inhibition of the reaction? What makes this sequence different from the d and e primers from part ? C. Diagrams on the left and right side are a bit hard to read- not sure how d to e or e to d happens. Need to make it clearer that nickase is playing a role in this reaction by splitting the primers after the polymerase makes the strand. D. This part is a lot better compared to the previous, but same thing as C, need to write in nickase for creating the output. Colloidal aggregation mechanism is clear and easy to understand.

Figure 2: A. Why is there no image of the aggregates? B. Image confirms that particles are assembling as expected. C. Hard to see whether image proves assembly is happening as hypothesized. There are pockets of green particles and pockets of red particles spread out in the image. May need to revise the aBy cluster image to show red clusters together as well which would be what happens if both particles cluster separately at first and then aggregate. Need more images to determine tunability of colloidal composition. Some method of quantifying amount of green vs red is also necessary to determine how time delay affects composition.

Figure 3: A. Images and cluster size graph do show that there is a clear localization of aggregation. B. Aggregation speed measured does appear proportional to concentration. Not sure how reaction rate is measured (concentration of d to e?). C. Graph convincingly shows d to e diffuses towards the end channel. D. Same here, diffusion is shown to occur compared to the to A. Bidirectionality of signal propagation is very interesting, maybe has a lot more applications than stated here.

Line 100: This equation suggests a square root relationship between the speed of the reaction and reaction rate, but Figure 3B has a graph that shows propagation speed to be linear with regards to template concentration. There needs to be a reference for the rate law of the reaction in order to determine if the speed hypothesis is correct. Like stated prior, need to relate rate to concentration.

Lines 133-135: "Kinetics and thermodynamics" needs far more explaining here rather than an assertion.

Figure 4: A. Scheme is fairly comprehensible but graphs could be titled to make it easier to tell what graph proves what. B. Needs an explanation for what happens at times i, ii, and iii, and how those cluster shapes prove that the clusters are disintegrating or aggregating at each time point. C. Cluster size graphs support oscillation. Bottom image needs time labels for each image and explanations for is happening at each snapshot.

Discussion: Very surprised at the brevity of this section. Propagation of signal and oscillation is quite interesting and applications should definitely be explored in this section. This reads more like a conclusion section. All the problems described in the abstract can be explored here in more detail with explanations on how this system is able to solve those problems and what advances in the field can occur due to solving those problems. This section must provide more context of the field and how this work will improve the field.

References: Should refer to previous literature more when putting these results into context as stated before. Weakest part of this manuscript is the lack of explanation for how these results are important. Lines 17-45 are well referenced and rest of the manuscript should aim to be similar to those lines.

Rebuttal Letter

Reviewer 1:

This manuscript describes mesoscopic particle assembly formation using enzymatic activity. The authors show dynamic control over the assembly of colloids by the use of variable DNA linker sequences. Results include oscillating colloidal aggregation over time, which is controllable by oscillating linker production over time. Overall, there is some really exciting science in this manuscript. Nevertheless, I have a real problem with this manuscript. The manuscript is poorly written and organized, and there are lots of typos, that make it hard to read and understand. It also bluntly misrepresents the field. For example the statement: 'Due to the lack of control and reversibility of the interactions, a dynamic and autonomous acting system on a mesoscale was not 10 feasible yet.' is just not true. There are a large number of papers in the literature that have used DNA melting temperature as a key part of a dynamic colloidal system. I understand what the authors want to say but the way they phrased it is misleading. Even colloidal machines (and I guess one can call them dynamic) have now been reported. Overall, this manuscript needs major revisions and clarification in some sections to clearly evince the study's points to readers. The manuscript is so full of jargon and unexplained abbreviations that even an expert on colloidal self-assembly has to re-read this manuscript multiple times to get the point.

We thank the reviewer for his/her detailed criticism. According to the suggestions we changed big parts of the introduction and rewrote several parts in the text to better explain the major points and put it into the perspective of the existing literature. We feel that the manuscript greatly benefited from this and that the accessibility for the broad range of readership is now improved significantly.

Check for typos. For example, tunable is spelled incorrectly in the abstract, production in figure 1 (german spelling), etc.

We thank the reviewer for this hint and corrected a number of typos and grammar issues.

Please re-think Figure 1. It would be great if you can provide a figure that shows the overall concept. Figure 1 is not it and if you think it is, it is WAY too complicated for the non-expert reader.

Reconsidering the Fig. 1 in the light of the raised criticism we fully agree with the reviewer and completely redrawn Figure 1. We redistributed the order of the figures and subfigures, added additional subtitles and explanations in the figure caption. We added a scheme to

explain the fundamental principles of our work, which explains the coupling of the enzymatic reaction network and the colloid interaction. In addition, we depicted the enzymatic reaction in more detail.

The Nomenclature is very confusing and never explained. For example, what is the difference between “ $\delta to \delta$ ” and “ $\delta to \delta$ ”, or they are just typos? More general, what is for example δ ? What is the DNA sequence (it is in the supplemental information but one cannot understand the article without it. This should be in the main text or at least the critical parts)? Why were these strands used? In Figure 2, there are spaces between “ $\delta to \delta$ ” (and other labels) that are not consistent with the rest of the work. The reader suggests, if possible, a change to “ $\delta to \delta$ ” (and similarly to all other labels) for the sake of readability and clarity. Moreover, the authors used “ $\delta to \alpha \beta$ ” to indicate linker-production, while use the say way of “ $\delta to \delta$ ” to represent the template for the production. Please find a clearer way to abbreviate the products or methods and make them consistent both in text and Figures.

We modified the abbreviations and made it more consistent and we added an explanation for our nomenclature in the main text to simplify the understanding of our abbreviations. All complementary strands and templates are now overlined. The abbreviations “ $\delta to \delta$ ” and “ $\delta to \delta$ ” are identical. We used the “space” in our figures only for a better illustration. We removed these differences and thank the author for this helpful note. In addition, we added the sequence of δ and ϵ to the main text and added an explanation for the choice of the used sequences.

The introduction should be edited to make clearer the background, motivation, and the proposed experimental methods of this work. For example, it is unclear how the work will relate to biological systems, which is the first topic introduced, until enzymatic activity is discussed 2 paragraphs later. It also does not provide a clear background of the state of the art in colloidal science. For example, self-replication has been reported for colloids. Cite it. Communication has been reported for colloidal systems. Same is true for inhibition. Same is true for dynamic switching.

We rewrote the introduction and improved the background and motivation of our work. In addition, we described and cited previous works within the field of colloidal research to give a clearer picture and put our work into the perspective of the existing literature.

Citations are missing in the introduction. Some sentences that introduce new concepts lack citations.

We added citations within the introduction, which describe the state of the art of colloidal science as well as the concept of using reaction networks to guide colloidal interactions.

Please always use either “past tense” or “present tense” when describing the experiments in your discussion part (either is fine but be consistent).

We completely rewrote the conclusion part and made it more consistent.

Figure 2. First, there should be a confocal image for illustration of $\alpha\beta$. Second, what does the yellow color indicate, since there are only green and red particles aggregating? Third, both confocal images in b and c are low quality and low magnification that it is not convincing to show the colloidal structure as described by cartoon images. One can actually see green-green interactions and red-red interactions. Quantification is needed.

We added a confocal image to illustrate the $\alpha\beta$ aggregation. The yellow colored particles of Fig. 2 b are in reality green particles. We changed the color-codes to prevent any misconceptions. We improved the contrast of our figures and zoomed into Fig.3 c to demonstrate the “green backbone” of the colloidal cluster.

In figure 4a, graphs are overlapping and should be moved for readability. For the figures used for explanation of tuning frequency with different concentrated $\delta t o \delta$, the ones shown in Figure S4c is less confusing than the current one

We change the Figure from S4c and the main figure to prevent the overlap and to show the change of frequency in a clearer way.

Figure 4 b, what are those three microscopy images (i, ii, iii)? There are no information or discussion about them.

I,II,III are microscopy snapshots at different points in time, indicated by the red circles. We added the information in the caption

Figure 4c, for $\delta t o \delta = 400\text{nM}$ group, as shown in the graph above, the cluster size at time 4.5-5 h was much smaller than the very beginning (individual particles). However, the clusters (around that time) look bigger and concentrated from the microscopy snapshots below

The experiments performed for the boxplots were realized using a thermocycler at constant working temperature and multiple identical samples. They were transferred to the heated observation chambers in 30 min time-steps. The length of the time-steps was chosen in order to effectively “scan” the oscillating profile of the DNA. Thereby, the oscillating

behaviour for the different template concentrations can be observed. However small differences occur because of the fixed time-steps.

The clusters of the microscopy snapshots of the video become bigger, since sedimentation plays an important role over such long time periods. The sedimentation leads to an upconcentration of the colloids, which can be seen by comparing the monodisperse colloids at different points in time.

Both experiments are described in the methods part (Microscopy experiments), which is now highlighted in the caption.

The discussion section reads as a conclusion.

We rewrote this part and described our results in a clearer way. We added a discussion part and highlighted the features and limitation of our experiments in order to give a perspective for future experiments.

Reviewer #2

We thank the reviewer 2 for the detailed criticism and suggestions. Addressing all of them helped to improve the manuscript considerably.

Abstract:

Need specifics on what material properties are accessible using the building blocks and interactions.

We added several examples of how colloidal building blocks and their interactions affect the material properties of a given system.

Unclear how DNA reaction circuits control the interactions and reversibility of particle assembly.

We added a half sentence to the abstract. The DNA reaction networks are able to produce DNA strands, which are able to bind DNA coated particles and induce their aggregation. The degradation of the produced DNA as well as DNA strand displacement reaction can lead to a disintegration of the particles.

A. "Figure 1: a. Figure is quite confusing. Hard to differentiate between how e and d mechanisms differ. **B.** How does the short polynucleotide sequence result in the inhibition of the reaction? What makes this sequence different from the d and e primers from part ? **C.** Diagrams on the left and right side are a bit hard to read- not sure how d to e or e to d happens. Need to make it clearer that nickase is playing a role in this reaction by splitting the primers after the polymerase makes the strand. **D.** This part is a lot better compared to the previous, but same thing as C, need to write in nickase for creating the output. Colloidal aggregation mechanism is clear and easy to understand."

A: "Figure 1: a. Figure is quite confusing. Hard to differentiate between how e and d mechanisms differ."

We remade Figure 1. We redistributed the order of the figures and subfigures, added additional subtitles and explanations in the figure caption. The concept of enzymatic reactions controlling the colloidal structure formation is now depicted in the first place for a better overview.

The mechanisms of the autocatalytic reactions of δ and ε are the same. Only the sequence of δ and ε and their template strands differ. Both reactions were designed and used, since they offer the opportunity to trigger different reactions and increase the variety of potential reactions and applications, like demonstrated for the communication. To make this clearer, we added the sequences of the primer strands in the main text.

B: How does the short polynucleotide sequence result in the inhibition of the reaction? What makes this sequence different from the d and e primers from part ?

The inhibition of the δ and ε reaction can be realised, by the elongation of the δ and ε with certain sequences, which are not complementary to the template strands. Here, the polymerase is no longer capable to elongate these strands and induce the autocatalytic reaction or linker production. This effect can be seen in the transformation reaction (predator: η) or within the inhibition reaction (poly t elongation).

C. Diagrams on the left and right side are a bit hard to read- not sure how $\delta\epsilon$ or $\epsilon\delta$ happens. Need to make it clearer that nickase is playing a role in this reaction by splitting the primers after the polymerase makes the strand.

We improved Fig.1 and depicted the enzymatic reaction process in more detail to demonstrate the interplay of the polymerase and nickase. Moreover, an improved explanation of the $\delta\epsilon$ and $\epsilon\delta$ reaction is now given in Fig 3.

D. This part is a lot better compared to the previous, but same thing as **C**, need to write in nickase for creating the output. Colloidal aggregation mechanism is clear and easy to understand.

The reaction of the linker production is depicted in more detail, including the nickase reaction.

Figure 2: **A.** Why is there no image of the aggregates? **B.** Image confirms that particles are assembling as expected. **C.** Hard to see whether image proves assembly is happening as hypothesized. There are pockets of green particles and pockets of red particles spread out in the image. May need to revise the aBy cluster image to show red clusters together as well which would be what happens if both particles cluster separately at first and then aggregate. Need more images to determine tunability of colloidal composition. Some method of quantifying amount of green vs red is also necessary to determine how time delay affects composition.

We included a picture of green aggregates and improved the quality of Fig. 2B. In addition, we increased the magnification factor of Fig 2C to show the green backbone of the aggregates. This experiment was realised by generating a time-delay of the $\alpha\gamma$ linker production in the presents of $\bar{\alpha}$, $\bar{\beta}$ and $\bar{\gamma}$ particles. The graph in Fig.2C, showing the controllable time-delay, was realised by varying the concentration of $\overline{\delta\epsilon}$ in order to control

the start of the $\overline{\varepsilon\tau o\alpha\gamma}$ linker production. This is demonstrated by the delayed aggregation of $\overline{\alpha}$ and $\overline{\gamma}$ particles for low $\overline{\delta\tau o\varepsilon}$ concentrations. In this experiment, we did not use β particles, in order to focus exclusively on the time-delay. As the reviewer correctly notes, one would expect a continuous shift from the hetero-aggregation to the “green-backbone” – clusters with increasing time-delay in the presence of all linker-templates and particles. We added a comment in the figure caption to clarify, that exclusively $\overline{\alpha}$ and $\overline{\gamma}$ aggregates were observed in this experiment.

Figure 3: **A.** Images and cluster size graph do show that there is a clear localization of aggregation. **B.** Aggregation speed measured does appear proportional to concentration. Not sure how reaction rate is measured (concentration of $\delta\tau o\delta$?). **C.** Graph convincingly shows $\delta\tau o\delta$ diffuses towards the end channel. **D.** Same here, diffusion is shown to occur compared to the to **A.** Bidirectionality of signal propagation is very interesting, maybe has a lot more applications than stated here.

The enzymatic reaction speed of the δ production was measured in bulk by fluorescent spectroscopy using different $\overline{\delta\tau o\delta}$ concentrations. The same $\overline{\delta\tau o\delta}$ concentrations were used to observe the tuneable propagation of δ and the corresponding aggregation. We changed the label of our x-axes from “Template” to “ $\overline{\delta\tau o\delta}$ concentration” to avoid any misconceptions and thank the reviewer for this hint.

Line 100: This equation suggests a square root relationship between the speed of the reaction and reaction rate, but Figure 3B has a graph that shows propagation speed to be linear with regards to template concentration. There needs to be a reference for the rate law of the reaction in order to determine if the speed hypothesis is correct. Like stated prior, need to relate rate to concentration.

The equation in line 100 describes the propagation of a compound, which is driven by “passive” diffusion and an “active” reaction. Such a reaction driven propagation results in a constant propagation-speed and exceeds the speed of passive diffusion, especially over long distances. Such a constant propagation for a reaction driven DNA network was already shown in [24]. There, one would expect a square root relationship between the speed of the reaction and the reaction rate. In our scenario, the reaction driven propagation of δ is not directly observed, since we image the aggregation of colloids. The observed aggregation wave additionally depends on the linker production and is delayed by the diffusion limited aggregation of the particles. Moreover, the $\overline{\delta\tau o\delta}$ reaction is a nonlinear reaction with an autocatalytic production and a negative feedback. Therefore, the linear increase of $\overline{\delta\tau o\delta}$ does not result in a linear increase of the reaction speed.

We understand, that the formula in line 100 can be misleading and one would expect the same relation for the colloidal aggregation. Therefore, we removed the equation in the main article and maintained with the quantitative explanation of the propagation.

Lines 133-135: "Kinetics and thermodynamics" needs far more explaining here rather than an assertion.

We added additional explanations, describing the kinetics and thermodynamics of the system and how they can be changed to tune the oscillations of our reaction network.

Figure 4: A. Scheme is fairly comprehensible but graphs could be titled to make it easier to tell what graph proves what. B. Needs an explanation for what happens at times i, ii, and iii, and how those cluster shapes prove that the clusters are disintegrating or aggregating at each time point. C. Cluster size graphs support oscillation. Bottom image needs time labels for each image and explanations for is happening at each snapshot.

We improved Fig. 4 and added titles to the subfigures for a better overview. In addition, we added an explanation for "i, ii and iii" and added the time labels of the colloidal clusters to the figure caption.

Discussion: Very surprised at the brevity of this section. Propagation of signal and oscillation is quite interesting and applications should definitely be explored in this section. This reads more like a conclusion section. All the problems described in the abstract can be explored here in more detail with explanations on how this system is able to solve those problems and what advances in the field can occur due to solving those problems. This section must provide more context of the field and how this work will improve the field.

We completely rewrote the discussion/conclusion part in order to put our results in a better context. In addition, we focussed on the main achievements of our work and described the challenges and features of our system in order to define the area of possible applications.

References: Should refer to previous literature more when putting these results into context as stated before. Weakest part of this manuscript is the lack of explanation for how these results are important. Lines 17-45 are well referenced and rest of the manuscript should aim to be similar to those lines.

We added several citations of the state of the art in colloidal science and refer to systems, which already used the concept of dynamic reaction networks to tune the colloidal interactions, in order to put our work in a better context. Moreover, we improved the introduction and discussion part for a better classification of our work and to highlight the main results and how they can be used for future applications.

REVIEWERS' COMMENTS

Reviewer #1 (Remarks to the Author):

The manuscript has improved significantly and now is a very nice contribution to the community. Figure 1a makes a huge difference to the understanding of the manuscript. There are a couple of minor points which I imagine Nature's typeset editor will address. Some spelling is US (functionalized) while others are in British spelling (organization). Also, I encourage the authors to remove 'novel' from the manuscript. My view is that all reported science should be 'new' and 'novel'. I still think that the images in Figure 2 are very low resolution. The scale bars at 10 microns and I think that they could be higher in resolution. Otherwise, the manuscript looks great.

Reviewer #2 (Remarks to the Author):

Manuscript can be accepted after grammar has been corrected.

Rebuttal Letter

Reviewer 1:

The manuscript has improved significantly and now is a very nice contribution to the community. Figure 1a makes a huge difference to the understanding of the manuscript. There are a couple go moonier points which I imagine Nature's type set editor will address. Some spelling is US (functionalized) while others are in british spelling (organization). Also, I encourage the authors to remove 'novel' from the manuscript. My view is that all reported science should be 'new' and 'novel'. I still think that the images in Figure 2 are very low resolution. The scale bars at 10 microns and I think that they could be higher in resolution. Otherwise, the manuscript looks great.

We thank the reviewer for his/her detailed criticism and his positive feedback. As the reviewer suggested, we removed „novel“ from the manuscript. Figure 2 shows a „zoom-in“ of our images, which explains our resolution. Since the overall concentration of colloids is comparable high we did not manage to get a better contrast/resolution.

Reviewer #2:

Manuscript can be accepted after grammar has been corrected.

We thank the reviewer for his comment and improved the grammar of our manuscript.